# Emergence and Evolution of OXA-23-Producing ST46_Pas_-ST462_Oxf_-KL28-OCL1 Carbapenem-Resistant *Acinetobacter baumannii* Mediated by a Novel IS*Aba1*-Based Tn*7534* Transposon

**DOI:** 10.3390/antibiotics12020396

**Published:** 2023-02-16

**Authors:** Haiyang Liu, Xiaochen Liu, Jintao He, Linghong Zhang, Feng Zhao, Zhihui Zhou, Xiaoting Hua, Yunsong Yu

**Affiliations:** 1Department of Infectious Diseases, Sir Run Run Shaw Hospital, Zhejiang University School of Medicine, Hangzhou 310016, China; 2Key Laboratory of Microbial Technology and Bioinformatics of Zhejiang Province, Hangzhou 310016, China; 3Regional Medical Center for National Institute of Respiratory Diseases, Sir Run Run Shaw Hospital, Zhejiang University School of Medicine, Hangzhou 310016, China; 4Department of Clinical Laboratory, Sir Run Run Shaw Hospital, Zhejiang University School of Medicine, Hangzhou 310016, China; 5Key Laboratory of Precision Medicine in Diagnosis and Monitoring Research of Zhejiang Province, Hangzhou 310016, China

**Keywords:** CRAB, ST46_Pas_, evolution, novel transposon, Tn*7534*, OXA-23

## Abstract

Carbapenem-resistant *Acinetobacter baumannii* (CRAB) isolates of global clone 1 (GC1) and global clone 2 (GC2) have been widely reported. Nevertheless, non-GC1 and non-GC2 CRAB strains have been studied less. In particular, no reports concerning sequence type 46 (ST46_Pas_) CRAB strains have been described thus far. In this work, the genomic features and possible evolution mechanism of ST46_Pas_ OXA-23-producing CRAB isolates from clinical specimens are reported for the first time. Antimicrobial susceptibility testing of three ST46_Pas_ strains revealed identical resistance profiles (resistance to imipenem, meropenem, ciprofloxacin and the combination of cefoperazone/sulbactam at a 2:1 ratio). They were found to belong to ST46_Pas_ and ST462_Oxf_ with capsular polysaccharide 28 (KL28) and lipooligosaccharide 1 (OCL1), respectively. Whole-genome sequencing (WGS) revealed that all contained one copy of chromosomal *bla*_OXA-23_, which was located in a novel IS*Aba1*-based Tn*7534* composite transposon. In particular, another copy of the Tn*7534* composite transposon was identified in an Hgz_103-type plasmid with 9 bp target site duplications (TSDs, ACAACATGC) in the *A. baumannii* ZHOU strain. As the strains originated from two neighboring intensive care units (ICUs), ST46_Pas_ OXA-23-producing CRAB strains may have evolved via transposition events or a p*dif* module. Based on the GenBank database, ST46_Pas_ strains were collected from various sources; however, most were collected in Hangzhou (China) from 2014 to 2021. Pan-genome analysis revealed 3276 core genes, 0 soft-core genes, 768 shell genes and 443 cloud genes shared among all ST46_Pas_ strains. In conclusion, the emergence of ST46_Pas_ CRAB strains might present a new threat to healthcare settings; therefore, effective surveillance is required to prevent further dissemination.

## 1. Introduction

*Acinetobacter baumannii* is an important pathogen that emerged only several decades ago, causing severe nosocomial infections due to its high-level resistance to various antimicrobial compounds, including carbapenems [1,2]. In 2019, the Centers for Disease Control and Prevention (CDC) recognized carbapenem-resistant *Acinetobacter* as one of the “Urgent Threats” and a top priority because of the limited options for treatment [3]. Carbapenem-resistant *A. baumannii* (CRAB) strains pose a tremendous global health issue, especially for hospitalized patients with immune dysfunction in intensive care units (ICUs) [4].

Research to date has established that the majority of CRAB isolates belong to global clone 1 (GC1) and global clone 2 (GC2), with GC2 being the most widespread clone worldwide [5]. Carbapenem resistance in *A. baumannii* strains is commonly mediated by OXA-23 oxacillinase [2,6]. Based on previous reports, the *bla*_OXA-23_ gene is most frequently located in IS*Aba1*-based transposons, namely Tn*2006*, Tn*2008*, Tn*2009* and AbaR4-type resistance islands [2,7,8,9]. These transposons can embed into the chromosome or insert into plasmids via IS*Aba1*-mediated transposition events, during which the signature of 9 bp target site duplications (TSDs) is formed [7,10,11].

Thus far, clones that do not belong to GC1 or GC2 CRAB strains have neither been reported nor studied in any detail, with very few exceptions. Moreover, ST46_Pas_ *A. baumannii* has been a relatively rare clone until now. Only one investigation from 2019 in Germany reported a carbapenem-susceptible *A. baumannii* (CSAB) collected from an infected animal (a horse with conjunctivitis), which belongs to ST46_Pas_ *A. baumannii* [12]. Researchers only described the draft genome sequence of *A. baumannii* strain 161514. Based on the Oxford scheme, ST46_Pas_ *A. baumannii* strains could be assigned to ST462_Oxf_ and have a close genetic relationship with ST1333_Oxf_ strains. However, there have been no related studies concerning ST462_Oxf_ and ST1333_Oxf_ clones to date.

In the current study, we characterized genomic features and possible evolutionary events occurring in three ST46_Pas_ OXA-23-producing clinical CRAB isolates for the first time. The genetic environment of the *bla*_OXA-23_ gene was analyzed in detail using a combination of short-read Illumina and long-read MinION whole-genome sequencing (WGS). A novel IS*Aba1*-based transposon (Tn*7534*) or p*dif* module may play a key role in the resistance gene transfer of *bla*_OXA-23_ from the chromosome to a plasmid with an Hgz_103-type replicon. We also performed comparative genomics of all ST46_Pas_ strains available in public databases and conducted the pan-genome analysis of the clone for the first time.

## 2. Results

### 2.1. Antimicrobial Susceptibility Profiles, Resistance Determinants and Virulence Factors

We characterized the resistome of three strains, *A. baumannii* DETAB-C9, DETAB-P65 and ZHOU. Antimicrobial susceptibility testing (AST) revealed that DETAB-C9, DETAB-P65 and ZHOU isolates were all resistant to imipenem (8–32 mg/L), meropenem (16–64 mg/L), cefoperazone/sulbactam (2:1 ratio) (64 mg/L) and ciprofloxacin (>32 mg/L). The *A. baumannii* ZHOU strain possessed comparably high MICs to carbapenems. However, three strains were still susceptible to ceftazidime (4 mg/L), amikacin (8 mg/L), colistin (1 mg/L) and tigecycline (0.5–1 mg/L) and remained intermediate to cefoperazone/sulbactam at a 1:1 ratio (32 mg/L) (Table 1).

Four resistance genes were found in all three *A. baumannii* strains: *bla*_OXA-23_, *bla*_OXA-67_, *bla*_ADC-26_ and *ant(3′′)-IIa*. In addition, DNA gyrase GyrA was discovered to have an amino acid mutation in position 81 (Ser81Leu), which may contribute to fluoroquinolones’ resistance.

Several virulence factors were detected in the three *A. baumannii* strains, including *bauABCDEF*, *basABCDEFGHIJ* and *barAB* encoding acinetobactin for iron uptake, poly-β-1,6-N-acetyl-d-glucosamine (PNAG) encoding the gene cluster *pgaABCD*, the outer-membrane-protein-related gene *ompA*, *pbpG* encoding PbpG for serum resistance, *csu* operon encoding Csu pili and two-component regulatory system *bfmRS* involved in Csu expression. Additionally, the lipopolysaccharide (LPS)-related genes *lpxABC* and *lpxL* were also identified.

### 2.2. Transfer of Carbapenemase Resistance Determinants

Mating assays and chemical transformation were conducted to study the transferability of *bla*_OXA-23_. However, despite several attempts, no transconjugants and transformants were obtained, possibly indicating that the determinants are not readily transferred.

### 2.3. Multilocus Sequence Typing (MLST), Capsular Polysaccharide (KL) and Lipoolygosaccharide (OCL)

According to the Pasteur and Oxford MLST schemes, all three strains belong to ST46_Pas_ (*cpn60*-5, *fusA*-12, *gltA*-11, *pyrG*-2, *recA*-14, *rplB*-9, *rpoB*-14) and ST462_Oxf_ (*cpn60*-16, *gdhB*-59, *gltA*-31, *gpi*-142, *gyrB*-33, *recA*-40, *rpoD*-7).

To analyze the KL and OCL of strains, Bautype and *Kaptive* software were used. The strains DETAB-C9 and DETAB-P65 were found to contain KL28 and OCL1, matching the 100% coverage to the reference sequence with 97.41% and 98.75% identity, respectively. Likewise, the KL and OCL in *A. baumannii* ZHOU were identical to those in DETAB-C9 and DETAB-P65, with a nucleotide identity of 97.41% and 98.77%. Similarly, identical KL and OCL results were obtained using *Kaptive* and Bautype.

### 2.4. Chromosome Analysis of ST46_Pas_ Strains

A hybrid assembly of short reads and long reads was performed to generate the complete genome sequences of *A. baumannii* strains. All strains exhibited a circular chromosome with a size of approximately 4 Mb and a GC content of 39% (Table 2). The chromosome of the *A. baumannii* ZHOU isolate contained a copy of the novel IS*Aba1*-derived Tn*7534* transposon. Corresponding to 28 bp site-specific recombinases XerC/XerD (C/D) and XerD/XerC (D/C) (also named p*dif*), 20 p*dif* sites were found in the chromosome of DETAB-C9 and DETAB-P65 strains. However, only 14 p*dif* sites were identified in the *A. baumannii* ZHOU chromosome, and p*dif* 2 and p*dif* 3 sites at each side of the *bla*_OXA-23_ segment (Appendix A). In addition, no prophage regions were identified in the chromosome of the three clinical strains.

### 2.5. Genetic Analysis of Plasmids

We identified one plasmid each in DETAB-C9 and DETAB-P65, which we called pDETAB-C9-1 and pDETAB7b, respectively. Based on the analysis of replicons, they both belonged to the Aci6 type with a size of 73,444 bp (Table 2). A similar plasmid, a 72,233 bp Aci6-type plasmid (called pZHOU-2), was identified in the ZHOU isolate, with 33.50% GC content (Table 2). Interestingly, no resistance genes were found to be encoded in the Aci6-type plasmids (Figure 1A). In contrast to the other two strains, another Hgz_103-type plasmid was identified in the *A. baumannii* ZHOU isolate, which we named pZHOU-1. A second copy of the novel IS*Aba1*-based Tn*7534* transposon was found in this plasmid, which exhibited the typical 9 bp target site duplications (TSD; ACAACATGC) (Figure 1B).

### 2.6. Plasmid Comparison and the Evolution of ST46_Pas_ CRAB Mediated by a Novel Tn7534 Transposon

With 99.99% identity and 94% coverage, the plasmid pZHOU-1 was identical to the *A. baumannii* ZW85-1 plasmid ZW85p2 (GenBank accession CP006769), first isolated in 2013 in Beijing (Figure 2A,B). However, no *oriT* or mobile genetic elements (MGEs) were found in the pZHOU-1 plasmid.

We then attempted to verify the evolutionary hypothesis about how the OXA-23-producing ST46_Pas_ CRAB originated. In detail, in step 1, a complete Tn*2009* composite transposon harboring *bla*_OXA-23_ was inserted into the *A. baumannii* chromosome to form 9 bp TSDs (AAAATATTT) flanking both sides (Figure 2C). Following this event, another copy of IS*Aba1* interrupted the original IS*Aba1*, possibly contributing to the generation of IS*Aba1Δ* and the formation of the novel transposon Tn*7534* (Figure 2D). Eventually, the Tn*7534* composite transposon was likely to integrate into the chromosome (Figure 2E). At one point, which cannot be conclusively determined from the sequence data (in parallel, before or after the genomic integration), the transposon was inserted into the pZHOU1 plasmid (Figure 2F), causing an 11 bp deletion that led to the formation of the signature 9 bp TSD (ACAACATGC).

### 2.7. Phylogenetic Analysis of All ST46_Pas_ A. baumannii Strains from NCBI Database

To further analyze the characteristics of ST46_Pas_ *A. baumannii* strains, a query using the NCBI GenBank database was performed. We found sequences of eight other ST46_Pas_ *A. baumannii* strains. Based on the MLST type of the Oxford scheme, all strains, apart from *A. baumannii* strain 161514, belonged to ST462_Oxf_. According to the previously published data, *A. baumannii* strain 161514 belonged to ST462_Oxf_ as well; however, it exhibited a mutation in *gdhB* (Oxf_*gdhB*_59 A > T). We then reanalyzed sequences that showed the strain belonged to a new ST_Oxf_, namely ST2098_Oxf_. Isolates were collected in Germany or China from several sources, including rectal swabs, conjunctivitis, sputum, blood and ascites, between 2014 and 2021 (Figure 3A). The hosts were mainly human, but—as previously mentioned—one strain was isolated from a horse (Equus ferus caballus). All ST46_Pas_ *A. baumannii* strains, which were collected from patients in Hangzhou (China), contained the *bla*_OXA-23_ gene. In contrast, no *bla*_OXA-23_ gene was identified in *A. baumannii* strain 161514 isolated from the equine in Germany. Genetic relationship analysis showed a rather close genetic connection between the two strains, DETAB-C9 and DETAB-P65. Additionally, three other strains (AB_HZ_B30, AB_HZ_S30, AB_HZ_B28) collected from another hospital but the same city, Hangzhou, were closely related.

### 2.8. Pan-Genome and Single Nucleotide Polymorphisms (SNPs) Analysis

SNP analysis data illustrate the large diversity of ST46_Pas_ *A. baumannii* strains. The DETAB-C9 and DETAB-P65 strains were found to be closely related, with only a 16-SNP difference observed (Figure 3B). A total of 18 to 33 SNPs were found in the AB_HZ_B30, AB_HZ_S30 and AB_HZ_B28 strains. In terms of the number of SNPs, there were large differences in the *A. baumannii* strain 161514 isolated in Germany when compared to the other strains, which were first found in China.

The genomes of all available ST46_Pas_ *A. baumannii* strains were re-annotated. We then performed a gene presence/absence analysis. Bioinformatic data revealed 3276 core genes, 0 soft-core genes, 768 shell genes and 443 cloud genes (Appendix A). The presence/absence of genes were visualized with the phylogenetic tree (Appendix A).

## 3. Discussion

Hospital-acquired infections caused by CRAB strains pose a growing clinical problem that has become a concern worldwide [13,14,15]. CRAB strains can persist in the hospital environment outside of patients, while at infection sites, they can form biofilms, further complicating treatment. Especially in the ICU, accidental dissemination can lead to infections in immunocompromised patients admitted due to non-infection-related medical issues [16,17,18].

In this work, we describe the first complete genome sequences of ST46_Pas_ and ST462_Oxf_ strains, which were collected from ICU patients in China. In 2019, Wareth et al. described a single draft genome sequence of a CSAB isolate recovered from a horse with conjunctivitis in Germany [12]. To our knowledge, this study is the first comprehensive report of ST46_Pas_ CRAB strains that also explores possible evolutionary pathways, including the integration and dissemination of composite transposons.

The regions encoding genes for the biosynthesis of capsular polysaccharide (KL) and the lipooligosaccharide outer core (OCL) are potentially highly valuable markers for tracking closely related isolates and assisting epidemiological surveillance in the hospital setting [19]. A well-conducted study by Wyres et al. established that OCL1 (2086/3029, 68.87%) was the most common type, followed by OCL3 (272/3029, 8.98%) [19]. Consistent with this observation, we observed that our strains contained OCL1, to a high degree of confidence. Moreover, by analyzing the KL type of genome assemblies downloaded from the NCBI GenBank database, we found that the most common KL types were KL2 (713/2948, 24.2%), KL9 (343/2948, 11.6%) and KL22 (330/2948, 11.2%) [19]. Hence, *A. baumannii* DETAB-C9, DETAB-P65 and ZHOU strains all exhibited a relatively rare type, KL28.

Mobile genetic elements (MGEs), including insertion sequences (ISs), integrons (In) and transposons (Tn), play a crucial role in antimicrobial resistance gene (ARG) transfer among different kinds of pathogens or between the chromosome and plasmids in a strain [20,21]. Here, the IS*26* transposase, a member of the IS*6* family, has the remarkable capability to spread ARGs across many Gram-negative bacteria [22,23], especially *Klebsiella pneumoniae* [24] and *E. coli* [25]. However, in *A. baumannii*, IS*Aba1*-based composite transposons are associated with carbapenem resistance and their transfer [8]. Here, we delineated the possible evolutionary pathway of the genetic segment containing *bla*_OXA-23_. The segment harboring *bla*_OXA-23_ likely emerged and transferred via translocation events of the novel IS*Aba1*-related Tn*7534* transposon. Previous studies uncovered that site-specific recombinases XerC/XerD (C/D) and XerD/XerC (D/C) are able to mediate ARG transfer, including that of *tet*(39) [26], *bla*_OXA-24_ [27] and *bla*_OXA-58_ [1]. In our study, we identified two C/D and D/C p*dif* sites near the Tn*7534* transposon containing the *bla*_OXA-23_ gene. The DNA segment encompassing the C/D and D/C p*dif* sites, including the Tn*7534* transposon, possibly forms a p*dif* module. Based on this data, we inferred that another possible mechanism of the mobilization of the Tn*7534* transposon containing the carbapenemase gene *bla*_OXA-23_ was the recombinase proteins XerC and XerD. A previous study revealed that prophage regions occasionally contain different resistance genes in *A. baumannii* strains, especially for *bla*_NDM-1_ and *bla*_OXA-23_, demonstrating an important role of phages in the transfer of ARGs [28]. In contrast to this report, we were unable to identify prophage regions in our three clinical strains. Consequently, phage-mediated transduction of the *bla*_OXA-23_ carbapenem resistance gene is unlikely for the strains we investigated.

Of note is the finding that all ST46_Pas_ CRAB strains were isolated in Hangzhou (China), as far back as 2014. This illustrates that ST46_Pas_ CRAB strains have existed in China for at least 8 years. Fortunately, its spread to other locations in the world has not been reported thus far. Our comprehensive pan-genome analysis, which highlighted the similarity of the strains based on the shared 3276 core genes, also revealed the diversity of the isolates, which we assessed on the basis of the differences in SNPs.

While our study aimed to characterize ST46_Pas_ CRAB strains, there were also limitations to our work. One is that we previously found the *A. baumannii* ZHOU strain to possess a hypermucoviscous phenotype, for which the virulence level could be further assessed [10]. Likely more important is the fact that we failed to establish the transferability of the Hgz_103-type plasmid harboring *bla*_OXA-23_ within the Tn*7534* transposon via conjugation or chemical transformation. One possible reason for this is that a T4SS-related transfer (*tra*) system, *oriT* region and relaxase were not identified in this plasmid, explaining the failure of our conjugation experiments. With regard to the transformation experiments, the uptake of DNA fragments via natural transformation with fragments larger than 50 kb is a challenge for bacteria [29]. Moreover, co-mobilization with another conjugative plasmid or the transfer via outer membrane vesicles could be explored. The lack of rapid and frequent transfer via any of the above mechanisms might explain the locally contained presence of the strains.

## 4. Materials and Methods

### 4.1. Patient Information, Bacterial Isolation and Identification

DETAB-C9 was isolated from the sputum of an 85-year-old male patient in the fourth-floor ICU of our hospital in Hangzhou, China, on 22 September 2019. This patient underwent screening of an oral swab and rectal swab every week. DETAB-P65 was collected from the rectal swab on 8 October 2019 from the same patient. In brief, the rectal swab was placed in 2 mL of tryptic soy broth (TSB) with 0.1% sodium thiosulfate and then incubated at 37 °C for one day. Then, 20 µL of overnight culture was plated onto *Acinetobacter* sp. CHROMagar plates (CHROMagar, Paris, France), which were supplemented with 2 mg/L meropenem. After 24 h of static culture at 37 °C, a single colony was chosen according to its color (red) and morphology, then streaked onto a Mueller–Hinton (MH) agar plate (Oxoid, Hampshire, UK), incubated overnight at 37 °C. A single colony was selected. Isolate identification was performed via matrix-assisted laser desorption ionization–time of flight mass spectrometry (MALDI-TOF MS; bioMérieux, Marcy-l’Étoile, France) and further confirmed via 16S rRNA gene-based PCR and sequencing.

*A. baumannii* strain ZHOU was collected from ascites in the clinical laboratory during the routine diagnostic of another 63-year-old male patient in the third-floor ICU of the same hospital on 2 December 2021.

### 4.2. Antimicrobial Agent Susceptibility Testing

Antimicrobial susceptibility against imipenem, meropenem, ceftazidime, cefoperazone/sulbactam (1:1 ratio), cefoperazone/sulbactam (2:1 ratio), amikacin, ciprofloxacin, colistin and tigecycline was determined using the broth microdilution method according to Clinical Laboratory Standards Institute (CLSI) 2021 standards. In particular, susceptibilities of cefoperazone/sulbactam at 1:1 and 2:1 ratios were determined based on the MICs of cefoperazone (MICs ≤ 8 mg/L denoting susceptibility, 16–32 mg/L indicating intermediate and ≥64 mg/L denoting resistance) [30]. *Escherichia coli* ATCC 25922 was used as the quality control strain.

### 4.3. Conjugation and Chemical Transformation Experiments

To determine the transferable ability of the Hgz_103-type plasmid carrying *bla*_OXA-23_ in the *A. baumannii* ZHOU strain, conjugation experiments using a rifampicin-resistant derivative of *A. baumannii* ATCC 17978 as the recipient strain were performed using the film mating method [1,18]. Transconjugants were selected on MH agar plates containing rifampicin (50 mg/L) and imipenem (2 mg/L). Donor or recipient bacteria culture alone was used as the control. The identity of transconjugants was confirmed via PCR. Experiments were carried out in triplicate independently.

Chemical transformation assays were further performed when conjugation experiments failed. The plasmid-harboring *bla*_OXA-23_ gene was transferred into *E. coli* DH5α via chemical transformation with imipenem (2 mg/L) for selection [31].

### 4.4. Whole-Genome Sequencing (WGS) and Phylogenetic Analysis

Genomic DNA was extracted from *A. baumannii* DETAB-C9, DETAB-P65 and ZHOU using a Qiagen minikit (Qiagen, Hilden, Germany) in accordance with the manufacturer’s instructions, followed by WGS with the Illumina HiSeq platform (Illumina, San Diego, CA, USA) and the MinION (Nanopore, Oxford, UK) platform (Weishu, Zhejiang, China). De novo assembly of the short and long reads was constructed using Unicycler v0.4.8 [32]. Assembly sequence quality was checked through QUAST v. 5.0.2 [33]. Genome sequence was annotated using both National Center for Biotechnology Information Prokaryotic Genome Annotation Pipeline (NCBI-PGAP) (http://www.ncbi.nlm.nih.gov/genome/annotation_prok/ (accessed on 7 September 2022) and Prokka 1.14.0 [34]. Pasteur [35] and Oxford [36] MLST schemes were performed via PubMLST (https://pubmlst.org/ (accessed on 7 September 2022)). For antimicrobial resistance profiles, ABRicate v0.8.13 was utilized with the ResFinder database [37]. Point mutations concerning resistance were identified using the fIDBAC software [38]. Virulence factors were detected via the virulence factor database (VFDB) [39]. Insertion sequences (ISs) and transposons were identified using ISFinder and The Transposon Registry [40,41]. *oriT*finder was used to recognize the origin of transfers (*oriT*) and mobile genetic elements (MGEs) [42]. The capsular polysaccharide (K locus) and lipooligosaccharide (OC locus) were analyzed by Bautype [43] and *Kaptive* [19], respectively. Recombinase recognition sites XerC/XerD (C/D) and XerD/XerC (D/C) were tested using p*dif*Finder (http://pdif.dmicrobe.cn/pdif/home/ (accessed on 7 September 2022)) [44]. The PHAge Search Tool (PHASTER) was utilized for bacteriophage prediction [45]. Plasmid structure was visualized with Proksee (https://proksee.ca/projects/new (accessed on 7 September 2022)). Plasmid comparison with ZW85p2 (accession number: CP006769) was performed using Easyfig [46] and visualized with Adobe Illustrator CC 2021. Phylogenetic analysis of all ST46 *A. baumannii* strains was performed using Snippy v4.4.5 (https://github.com/tseemann/snippy (accessed on 21 September 2022)) and FastTree using RAxML under the GTRGAMMA model with DETAB-C9 as the reference strain [47]. Generation tree file was visualized using iTOL v5 (https://itol.embl.de/ (accessed on 27 September 2022)) [48]. Single nucleotide polymorphisms (SNPs) were calculated using SNP-dists V0.6.3 (https://github.com/tseemann/snp-dists (accessed on 27 September 2022)) [49], and the matrix was further visualized with RStudio v3.5.3.

### 4.5. Pan-Genome Analysis

Pan-genome analysis of all ST46_Pas_ *A. baumannii* strains from NCBI GenBank Assembly database was conducted using the BacWGSTdb server to search the close genetic relationship strains with the parameter of 100 SNPs threshold [50]. In brief, Roary v3.11.2 pipeline [51] was used to perform the pan-genome analysis using the GFF files generated by Prokka, with a blastp percentage identity of 95% and a core definition of 99%. Based on the pan-genome analysis, four various classes of genes were classified, including “core” (99% ≤ strains ≤ 100%), “soft core” (95% ≤ strains < 99%), “shell” (15% ≤ strains < 95%) and “cloud” (0% ≤ strains < 15%) groups, respectively [52]. Gene presence/absence file was further visualized with the tree file using the phandango website (https://jameshadfield.github.io/phandango (accessed on 27 September 2022)).

### 4.6. Nucleotide Sequence Accession Numbers

The complete genome assemblies of the chromosome and plasmids from *A. baumannii* DETAB-C9, DETAB-P65 and ZHOU are available from GenBank under accession numbers CP104295-CP104296, CP077835-CP077836 and CP104297-CP104299, respectively.

## 5. Conclusions

This study describes, for the first time, the genomic characteristics of ST46_Pas_ OXA-23-producing CRAB isolates. Possible evolutionary pathways indicate that the strains emerged via translocation events of the IS*Aba1*-related Tn*7534* composite transposon or the p*dif* module. Thus far, ST46_Pas_ CRAB strains have been mainly recovered from hospitals in Hangzhou, China. Nonetheless, effective surveillance should be implemented for ST46_Pas_ CRAB strains to prevent their dissemination and outbreaks in China and globally.

## Figures and Tables

**Figure 1 antibiotics-12-00396-f001:**
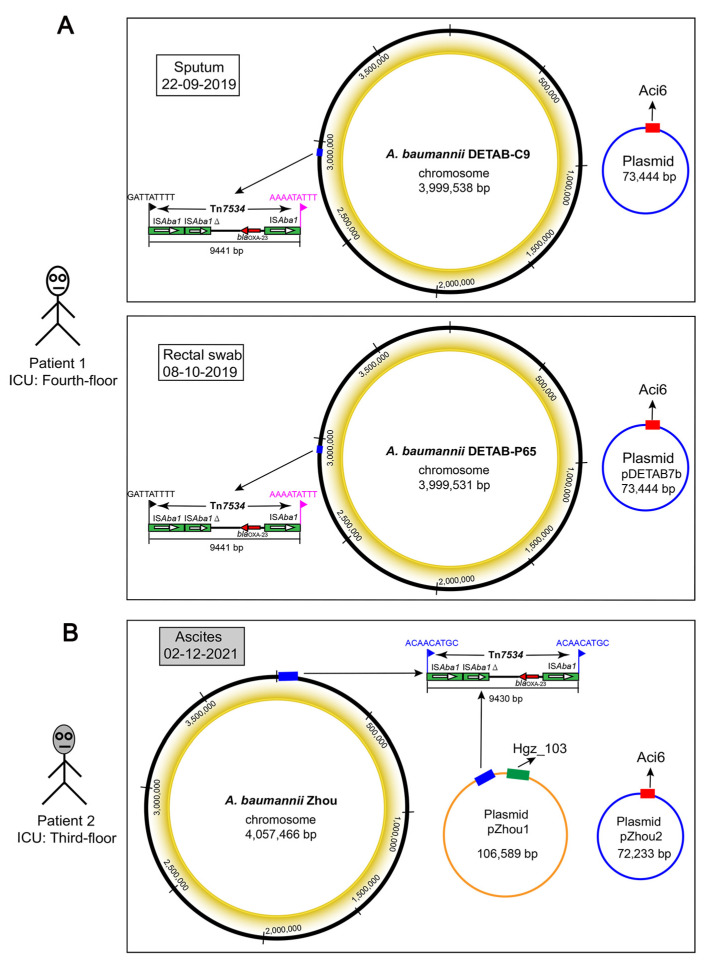
*A. baumannii* strains and genomic information. (**A**) *A. baumannii* DETAB-C9 and DETAB-P65 were collected from the sputum and rectal swab of patient 1 (upper white cartoon character) on the fourth floor of the ICU. Circular maps of the chromosome and plasmid are shown using black and blue circles, respectively. The position of novel transposon Tn*7534* is indicated by the blue box, and its specific structure is drawn with 9 bp target site duplications (TSDs) indicated using different-colored flags based on the TSD sequence. (**B**) The *A. baumannii* ZHOU strain was collected from ascites of patient 2 (lower grey cartoon character) on the third floor of the ICU in the same hospital. Circular maps of the chromosome and plasmid are also shown. The Hgz_103-type plasmid is indicated by an orange circle. Two copies of Tn*7534* transposon are indicated by a blue box with 9 bp identical TSDs (ACAACATGC) on either side.

**Figure 2 antibiotics-12-00396-f002:**
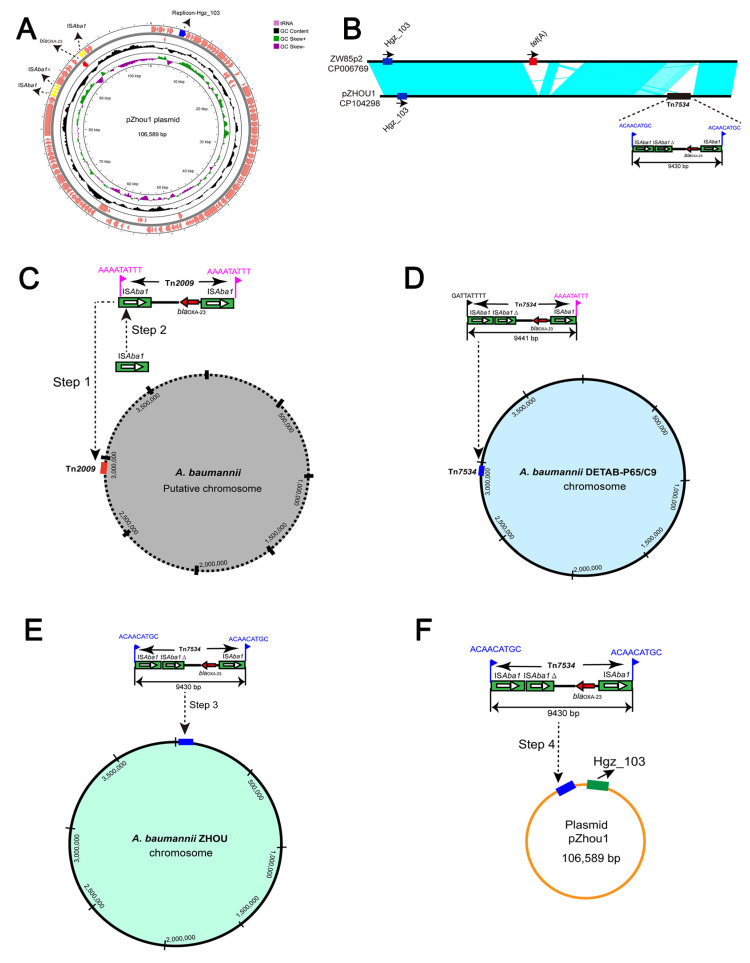
Circular map of Hgz-103-type plasmid pZhou1 and comparison with ZW85p2 and the proposed diagram for the evolution of ST46_Pas_ CRAB. (**A**) Circular map of Hgz-103-type pZhou1 plasmid. Arrows show the direction of ORFs. Red arrow indicates the *bla*_OXA-23_ gene. Blue arrow indicates the Hgz-103 replicon. Yellow arrows represent IS*Aba1*. (**B**) Linear maps and comparison with the ZW85p2 plasmid (accession number: CP006769). Blue arrows indicate the Hgz-103 replicons, with black arrows indicating the direction of ORFs. Red box indicates the resistance gene *tet*(A). The Tn*7534* transposon is shown and labeled with a 9 bp TSD (ACAACATGC). Homologous segments (representing ≥99% identity) are indicated by light blue shading. (**C**) The putative ST46_Pas_ chromosome circle map. Tn*2009* transposon is indicated by a red box with a 9 bp TSD (AAAATATTT). The *bla*_OXA-23_ gene is shown by the red arrow. IS*Aba1* is indicated by a green box with a white arrow. The complete Tn*2009* composite transposon was integrated into the putative *A. baumannii* chromosome, followed by another copy of IS*Aba1*, which interrupted the original IS*Aba1*, contributing to generating IS*Aba1Δ* to form transposon Tn*7534* in *A. baumannii* DETAB-C9 or DETAB-P65, as shown in (**D**). (**E**,**F**) The Tn*7534* composite transposon was integrated into the *A. baumannii* ZHOU chromosome and pZHOU1 plasmid. The 9 bp TSDs (ACAACATGC) are indicated by blue flags.

**Figure 3 antibiotics-12-00396-f003:**
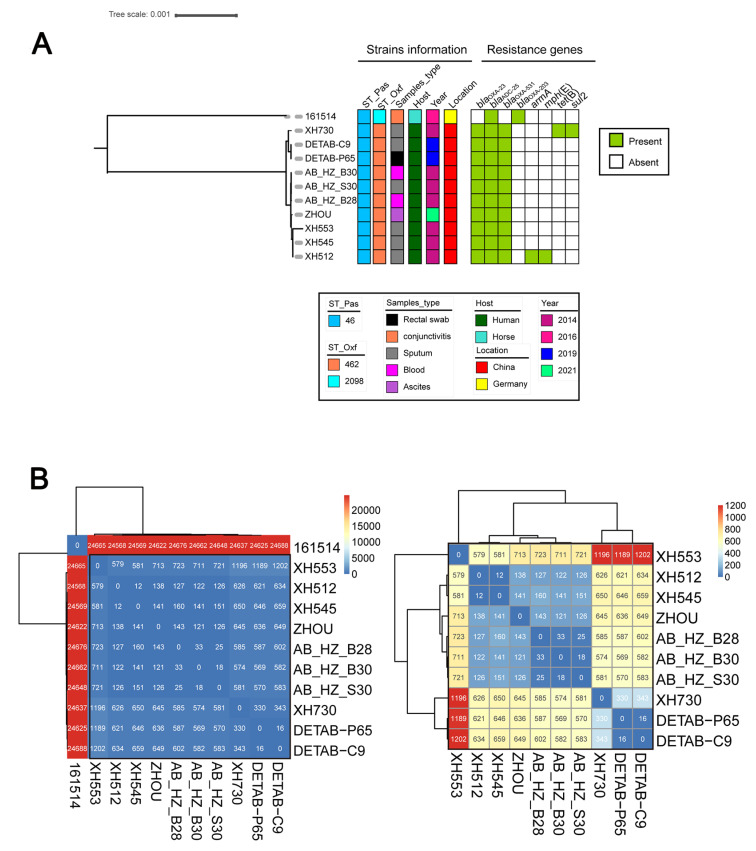
Phylogenetic analysis of 11 ST46_Pas_ *A. baumannii* strains and SNPs matrix values among ST46_Pas_ strains. (**A**) The tree was built with Snippy v4.4.5 and FastTree using RAxML under the GTRGAMMA model with DETAB-C9 as the reference strain and visualized with iTOL v5. Isolate name, Pasteur and Oxford MLST schemes, sample type, host, collection locations, isolation date and the heatmap of resistance genes are shown for each strain. The GenBank accession numbers of other ST46_Pas_ *A. baumannii* strains from the NCBI GenBank public database are as follows: 161514 (GenBank accession number: RPDK01000001), XH730 (GenBank accession number: LYHV01000001), AB_HZ_B30 (GenBank accession number: PRIR01000001), AB_HZ_S30 (GenBank accession number: PRFV01000001), AB_HZ_B28 (GenBank accession number: PRJO01000001), XH553 (GenBank accession number: LYKU01000001), XH545 (GenBank accession number: LYLC01000001), XH512 (GenBank accession number: LYLG01000001). (**B**) SNP matrix values among ST46_Pas_ strains. SNP differences are indicated as numbers in the boxes. Strains in the black square are the same.

**Table 1 antibiotics-12-00396-t001:** Minimum inhibitory concentrations of antibiotics used in this study.

Isolates	Antibiotics ^1^ Minimum Inhibitory Concentration (mg/L)								
	IMP	MEM	CAZ	SCF (1:1)	SCF (2:1)	AMI	CIP	COL	TGC
DETAB-C9	16	32	4	32	64	8	>32	1	1
DETAB-P65	8	16	4	32	64	8	>32	1	0.5
ZHOU	32	64	4	32	64	8	>32	1	0.5

^1^ IMP = imipenem, MEM = meropenem, CAZ = ceftazidime, SCF (1:1) = cefperazone/sulbactam (1:1 ratio), SCF (2:1) = cefoperazone/sulbactam (2:1 ratio), AMI = amikacin, CIP = ciprofloxacin, COL = colistin, TGC = tigecycline.

**Table 2 antibiotics-12-00396-t002:** Characteristics of genome components.

Isolates	Element	Replicon	Size (bp)	GC (%)	Antibiotic Resistance Genes
DETAB-C9	chromosome	ND	3,999,538	38.98%	*bla*_OXA-23_, *bla*_OXA-67_, *bla*_ADC-26_, *ant(3″)-IIa*
	pDETAB-C9-1	Aci6	73,444	33.40%	Not detected
DETAB-P65	chromosome	ND	3,999,531	38.98%	*bla*_OXA-23_, *bla*_OXA-67_, *bla*_ADC-26_, *ant(3″)-IIa*
	pDETAB7b	Aci6	73,444	33.40%	Not detected
ZHOU	chromosome	ND	4,057,466	39.02%	*bla*_OXA-23_, *bla*_OXA-67_, *bla*_ADC-26_, *ant(3″)-IIa*
	pZHOU-1	Hgz_103	106,589	41.38%	*bla* _OXA-23_
	pZHOU-2	Aci6	72,233	33.50%	Not detected

## Data Availability

The complete genome assemblies of the chromosome and plasmids from *A. baumannii* DETAB-C9, DETAB-P65 and ZHOU are available from GenBank under accession numbers CP104295-CP104296, CP077835-CP077836 and CP104297-CP104299, respectively. The data link for reviewers is https://www.ncbi.nlm.nih.gov/search/all/?term=PRJNA738868 (accessed on 17 June 2021) and https://www.ncbi.nlm.nih.gov/search/all/?term=+PRJNA877402 (accessed on 7 September 2022).

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
