# Peer review of "Emergence and Evolution of OXA-23-Producing ST46Pas-ST462Oxf-KL28-OCL1 Carbapenem-Resistant Acinetobacter baumannii Mediated by a Novel ISAba1-Based Tn7534 Transposon"

_antibiotics, 2023, doi:10.3390/antibiotics12020396_

Round 1

Reviewer 1 Report

The topic is very interesting e deserves to be considered for publication. CRAB nosocomial spreading represents one of the most complex challenge, and a detailed analysis of the mechanisms responsible for hospital transmission of this emergent pathogen is of huge importance. The genetic characterization using Whole genome sequencing (WGS) and the phylogenetic analysis are well-conducted and extensive allowing for a better understanding the possible dissemination mechanism of ST46Pas OXA-23-producing CRAB isolates. 

Author Response

Reviewer 1

The topic is very interesting e deserves to be considered for publication. CRAB nosocomial spreading represents one of the most complex challenge, and a detailed analysis of the mechanisms responsible for hospital transmission of this emergent pathogen is of huge importance. The genetic characterization using Whole genome sequencing (WGS) and the phylogenetic analysis are well-conducted and extensive allowing for a better understanding the possible dissemination mechanism of ST46Pas OXA-23-producing CRAB isolates.

**Thanks for your comments and the work of review!

Reviewer 2 Report

The authors described for the first time the genomic characteristics of STpas OXA-23 producing Carbapenem resistant Acinetobacter baumannii. This study is relevant from an epidemiological point of view, it is original and very well presented, I wish I always received papers of such quality. I have some comments to improve the overall quality.

Table 1: It is not necessary to indicate if the strain is resistant or susceptible. Please remove this information from the table.

The manuscript presents too many figures. My suggestion is to combine Figures 2 and 3 in only one (A-D), and Figures 4 and 5 in one. Figure 5 A and 5B could be moved to supplementary material, given the fact that the information that they show is clearly explained in the text.

The numbers in the Figure 5C are difficult to read, mainly in the red and blue squares.

In Figure 4, please indicate which methodology was used for the construction of the phylogenetic tree.

Regarding the text:

Lines 164-173: The paragraph should be written as a hypothesis, not as a fact.

Lines 266-269: The sentence is not clear. Please modify it.

Line 302: Please indicate the brand of the CHROMagar plates.

Reviewer 3 Report

The authors did a splendid job with the manuscript. I had not found any major concerns with the manuscript. This is a well written article with focused study. I have a couple of minor concerns. 

How was the search for  ST46Pas A. baumannii strains done in NCBI genbank database? What does the query entail? 

Introduction section needs expansion, especially in introducing ST46Pas sequence type. 

Author Response

Reviewer 3

The authors did a splendid job with the manuscript. I had not found any major concerns with the manuscript. This is a well written article with focused study. I have a couple of minor concerns.

How was the search for ST46Pas A. baumannii strains done in NCBI genbank database? What does the query entail?

**We selected these ST46Pas strains using BacWGSTdb server (http://bacdb.cn/BacWGSTdb/). BacWGSTdb 2.0 server is a free publicly accessible database which have developed for bacterial whole-genome sequence typing and source tracking (PMID: 33010178, Nucleic Acids Res. 2021 Jan 8;49(D1):D644-D650. doi: 10.1093/nar/gkaa821). When I uploaded the genome of one of my ST46Pas strain and set the parameter with 100 SNPs threshold, this server will help to find all strains with close genetic relationship, including all ST46Pas and provide the Genbank ID, then I could download them from Genbank and perform bioinformatic analysis later. I have added this detailed information in the manuscript of the “Method” section for pan-genome analysis.

Introduction section needs expansion, especially in introducing ST46Pas sequence type.

**Thanks for your suggestions. We have added some content in the “Introduction” section. Please see “As for ST46Pas A. baumannii, it is a quite rare clone until now. Only one study from 2019 in Germany reported a carbapenem-susceptible A. baumannii (CSAB) collected from an infected animal (a horse with conjunctivitis) which belongs to a ST46Pas A. baumannii. Researchers only described the draft genome sequence of A. baumannii strain 161514. Based on the Oxford scheme, ST46Pas A. baumannii strains could be assigned to ST462Oxf and have a near genetic relationship with ST1333Oxf strains. However, there are also no related studies concerning this ST462Oxf and ST1333Oxf clones to date.” in line 59-66.